# Inverse classification with logistic and softmax classifiers: efficient optimization

**Miguel Á. Carreira-Perpiñán**                                   *mcarreira-perpinan@ucmerced.edu*
*Dept. of Computer Science & Engineering*
*University of California, Merced*

**Suryabhan Singh Hada**[*]                                              *shada@paypal.com*
*PayPal*

**Reviewed on OpenReview:** *https://openreview.net/forum?id=ZrNhf7P3a1*

## Abstract

In recent years, a certain type of problems have become of interest where one wants to query a trained classifier. Specifically, one wants to find the closest instance to a given input instance such that the classifier's predicted label is changed in a desired way. Examples of these "inverse classification" problems are counterfactual explanations, adversarial examples and model inversion. All of them are fundamentally optimization problems over the input instance vector involving a fixed classifier, and it is of interest to achieve a fast solution for interactive or real-time applications. We focus on solving this problem efficiently with the squared Euclidean distance for two of the most widely used classifiers: logistic regression and softmax classifier. Owing to special properties of these models, we show that the optimization can be solved in closed form for logistic regression, and iteratively but extremely fast for the softmax classifier. This allows us to solve either case exactly (to nearly machine precision) in a runtime of milliseconds to around a second even for very high-dimensional instances and many classes.

## 1 Introduction

In an abstract sense, machine learning classifiers can be regarded as manipulating three objects: the input feature vector $\mathbf{x}$, the output label $y$ and the classifier model $f$. *Training* or *learning* $(\mathbf{x}, y \to f)$ is the problem of inferring $f$ from $\mathbf{x}$ and $y$ (the training set), and is usually formulated as an optimization problem. *Inference* or *prediction* $(\mathbf{x}, f \to y)$ is the problem of inferring $y$ from $\mathbf{x}$ and $f$ (the training set), and is usually formulated via the application of an explicit function $f$ to $\mathbf{x}$. Both training and inference have been long studied for a wide range of classifiers. The third problem $(f, y \to \mathbf{x})$, which can be called *inverse classification*, is to find $\mathbf{x}$ from $y$ and $f$, and can also be formulated as an optimization problem. This problem is far less popular than the other two but has received significant attention in recent years in specific settings. One of them is in *adversarial examples*. We are given a classifier $f$, such as a neural net, and an input instance $\mathbf{x}$ which is classified by $f$ as (say) class 1. The goal is to perturb $\mathbf{x}$ slightly (or equivalently to find a close instance to $\mathbf{x}$) such that it is then classified by $f$ as (say) class 2. The motivation is to trick the classifier into predicting another class and trigger some action (such as having a self-driving car misclassify a stop sign as a yield sign, say). Another setting is in *counterfactual explanations*. Again, we are given a classifier $f$ and an input instance $\mathbf{x}$ which is classified by $f$ as (say) class 1, which is undesirable. The goal is to change $\mathbf{x}$ in a minimally costly way so that it is classified as (say) class 2, which is desirable. As an example, $\mathbf{x}$ could represent a loan applicant (age, salary, etc.) and $f$ could predict whether to approve the loan or not.

---

[*]S. S. Hada's work took place while he was at UC Merced.

Problems of this type have become particularly important in the context of transparency in AI, where users may want to interpret, explain, understand, query or manipulate a trained classifier.

Such inverse classification problems can be naturally formulated as an optimization involving a distance or cost in feature space and a loss in label space. While many recent works have focused on the motivation or application of adversarial examples and counterfactual explanations, here we are interested in the numerical optimization aspects of such problems. Indeed, we recognize it as a novel type of optimization problem with specific characteristics that deserves special attention. Such problems are computationally much smaller than the training problem (which involves a dataset and model parameters as variables), but it is still important to solve them fast: some of the applications mentioned above require real-time or interactive processing, and this processing may often take place in limited-computation devices such as mobile phones. Besides, the optimization can be nontrivial if the number of features is in the thousands or milions.

In this paper, we consider logistic regression and softmax classifiers (also known as multinomial or multiclass logistic regression), which are among the simplest yet most widely used classifiers in practice; and the squared Euclidean distance as cost. We will define inverse classification as a certain optimization problem that, while encoding the idea of inverse classification in a natural way, is computationally convenient. Then, we characterize it in theory and provide what probably are the most efficient solutions possible for either type of classifier.

## 2 Related work

As noted earlier, some forms of inverse classification problems exist, although generally the emphasis has been in the application of such problems rather than on understanding the optimization problem in theory and solving it efficiently.

Most inverse classification works concern neural networks. Inverting neural nets with continuous outputs has been investigated since the 1980s (Williams, 1986; Kindermann & Linden, 1990; Hoskins et al., 1992; Behera et al., 1995; Jensen et al., 1999). The algorithms were usually based on minimizing the squared error between the desired output and the predicted one via backpropagation, i.e., gradient descent. The widespread deployment of large, highly accurate neural nets in many practical applications in the last decade has brought new interest in inversion problems. One of them is activation maximization, which—with the goal of understanding the inner workings of a deep net—seeks input vectors that cause a particular neuron in the net to have a large output (Simonyan et al., 2014; Zeiler & Fergus, 2014; Mahendran & Vedaldi, 2016; Dosovitskiy & Brox, 2016; Nguyen et al., 2016; Hada & Carreira-Perpiñán, 2021a). Another is adversarial examples, mentioned in the introduction (Szegedy et al., 2014; Goodfellow et al., 2015). Again, these works typically use gradient descent as optimization method, which is greatly facilitated by automatic differentiation, now commonly available in libraries such as PyTorch or Tensorflow (although the ReLU activation commonly used in deep nets is not differentiable everywhere). Some of these works also explore different formulations of the problem, such as different distance or cost functions, different constraints or different ways to perturb the instance, sometimes in an effort to find input instances that are realistic or suitable for specific data, such as images, audio or language (Nguyen et al., 2017; Dhurandhar et al., 2018; Alzantot et al., 2018; Carlini & Wagner, 2018; Goodfellow et al., 2018; Finlayson et al., 2019; Van Looveren & Klaise, 2021).

A more recent trend are counterfactual explanations (Wachter et al., 2018; Ustun et al., 2019; Russell, 2019; Karimi et al., 2020). Some of these use agnostic algorithms which assume nothing about the classifier other than it can be evaluated on arbitrary instances, using some kind of random or approximate search (Sharma et al., 2020; Karimi et al., 2020). One can also restrict the instance search space to a finite set of instances and do a brute-force search, as in a database search (Wexler et al., 2020). Other algorithms are specialized for decision trees and forests, which have the added difficulty of defining a piecewise constant classifier or regressor (Kantchelian et al., 2016; Chen et al., 2019; Parmentier & Vidal, 2021; Carreira-Perpiñán & Hada, 2021a;b; Hada & Carreira-Perpiñán, 2021b; Carreira-Perpiñán & Hada, 2023a;b).

Driven by practical applications, problems related to inverse classification have also been studied in data mining or knowledge discovery from databases as a form of knowledge extraction from a trained model

(Yang et al., 2006; Bella et al., 2011; Martens & Provost, 2014; Cui et al., 2015), in particular in applications such as customer relationship management (CRM).

Finally, inverse classification or regression arises also as a step within the method of auxiliary coordinates (MAC) (Carreira-Perpiñán & Wang, 2012; 2014). MAC seeks to train nested functions in machine learning, resulting from function composition, such as $\mathbf{g}(\mathbf{f}(\mathbf{x}))$ where $\mathbf{f}$ is a feature extraction stage whose output is fed to a classifier $\mathbf{g}$. MAC operates by introducing auxiliary variables $\mathbf{z}$ that decouple $\mathbf{f}$ from $\mathbf{g}$, turning this into a penalized optimization, and solving it by alternating over the original parameters of $\mathbf{f}$ and $\mathbf{g}$ and the auxiliary variables $\mathbf{z}$. The latter step over the auxiliary variable $\mathbf{z}_n$ of each training instance $(\mathbf{x}_n, \mathbf{y}_n)$ can be written as "$\min_{\mathbf{z}_n} \frac{\lambda}{2}\|\mathbf{z}_n - \mathbf{f}(\mathbf{x}_n)\|^2 + \mathrm{loss}(\mathbf{y}_n, \mathbf{g}(\mathbf{z}_n))$", which takes the form of an inverse classification or regression, similar to eq. (2).

## 3 Inverse classification as optimization: softmax classifier

### 3.1 Definition of the optimization problem

Consider a $K$-class softmax classifier, where the softmax probability of class $i \in \{1, \ldots, K\}$ for instance $\mathbf{x} \in \mathbb{R}^D$ is defined as $\mathbf{p}(\mathbf{x}) = (p_1(\mathbf{x}), \ldots, p_K(\mathbf{x}))^T$ with

$$p_i(\mathbf{x}) = e^{z_i} \Big/ \sum_{j=1}^{K} e^{z_j}, \ i \in \{1, \ldots, K\} \tag{1}$$

where $\mathbf{z} = \mathbf{A}\mathbf{x} + \mathbf{b} \in \mathbb{R}^K$ and the parameters $\mathbf{A} \in \mathbb{R}^{K \times D}$ and $\mathbf{b} \in \mathbb{R}^{D \times 1}$ are a matrix of weights and a vector of biases, respectively. Throughout this paper, we assume the parameters (hence the classifier) are fixed, presumably by having been learned on a training set. Obviously, each $p_i$ value is in (0,1) and their sum is 1. Define the function $g_i(\mathbf{x}) = -\ln p_i(\mathbf{x}) > 0$. Write $\mathbf{A}^T = (\mathbf{a}_1 \cdots \mathbf{a}_K)$ where each $\mathbf{a}_i$ is of $D \times 1$ ($i$th row of $\mathbf{A}$, transposed). Define $\bar{\mathbf{A}}_k = \mathbf{A} - \mathbf{1}\mathbf{a}_k^T$ of $K \times D$, where $\mathbf{1}$ is a vector of ones (and likewise $\bar{\mathbf{a}}_{ki} = \mathbf{a}_i - \mathbf{a}_k$) and $\bar{\mathbf{b}}_k = \mathbf{b} - b_k\mathbf{1}$ of $K \times 1$. Note that $\bar{\mathbf{A}}_k$ has one row of zeros (row $k$).

The following results characterize the problem. Theorems 3.1–3.2 are well known in convex optimization. The remaining theorems exploit the particular mathematical structure of the $K$-class softmax classifier, logistic regression and the $\ell_2$ norm (the low-dimensional Hessian representation for the $K$-class softmax classifier and the closed-form solution for logistic regression). This affords important computational advantages which do not arise with other classifiers or other norms.

**Theorem 3.1.** *For each $i \in \{1, \ldots, K\}$ we have that $g_i$ is convex (but not strongly convex).*

*Proof.* This is a well-known result, see e.g. Boyd & Vandenberghe (2004, p. 87). $\qquad\square$

We define the optimization problem[1] [2]:

$$\min_{\mathbf{x} \in \mathbb{R}^D} E(\mathbf{x}; \lambda, k) = \frac{\lambda}{2}\|\mathbf{x} - \overline{\mathbf{x}}\|^2 + g_k(\mathbf{x}) \tag{2}$$

where $\overline{\mathbf{x}} \in \mathbb{R}^D$ is the *source instance*, $k \in \{1, \ldots, K\}$ the *target class* and $\lambda > 0$ trades off distance to $\overline{\mathbf{x}}$ with class-$k$ probability. Next, we give several results about the problem and about Newton's method and discuss their implications in section 3.6. An alternative formulation of interest is "$\min_{\mathbf{x}} \|\mathbf{x} - \overline{\mathbf{x}}\|$ s.t. $g_k(\mathbf{x}) \leq \alpha$". Although we do not consider it here, it can be solved by trying different values of $\lambda$ in (2), possibly using a bisection search. This can be done efficiently by scanning a solution path over $\lambda$ via warm-start (see section 3.5).

**Theorem 3.2.** *$E(\mathbf{x}; \lambda, k)$ is strongly convex. Hence, it has a unique minimizer $\mathbf{x}^*$.*

---

[1]All (vector or matrix) norms are $\ell_2$ norms throughout the paper.

[2]Note that $-p_k$ itself is not convex (even for $K = 2$) and neither is $\frac{\lambda}{2}\|\mathbf{x} - \overline{\mathbf{x}}\|^2 - p_k(\mathbf{x})$. Defining a model inversion this way has local minima (in fact, it seems to have a local minimum very near $\overline{\mathbf{x}}$, which is useless but makes finding a useful minimum hard).

*Proof.* This follows (Nesterov, 2004) from the fact that $E$ equals the sum of a convex function, $g_k$ (from theorem 3.1), and a strongly convex quadratic function, $\frac{\lambda}{2}\|\mathbf{x} - \overline{\mathbf{x}}\|^2$. □

### 3.2 Gradient and Hessian

It is easy to see that the gradient and Hessian of $E$ wrt $\mathbf{x}$ are:

$$\nabla_{\mathbf{x}} E(\mathbf{x}; \lambda, k) = \lambda(\mathbf{x} - \overline{\mathbf{x}}) + \bar{\mathbf{A}}_k^T \mathbf{p}(\mathbf{x}) \tag{3}$$

$$\nabla_{\mathbf{xx}}^2 E(\mathbf{x}; \lambda, k) = \lambda\mathbf{I} + \bar{\mathbf{A}}_k^T (\operatorname{diag}(\mathbf{p}) - \mathbf{pp}^T)\bar{\mathbf{A}}_k. \tag{4}$$

To avoid clutter, we will usually omit the dependence on $\mathbf{x}, \lambda, k$. We will also call the matrix $\bar{\mathbf{A}}_k^T(\operatorname{diag}(\mathbf{p}) - \mathbf{pp}^T)\bar{\mathbf{A}}_k$ the "softmax Hessian" .

**Theorem 3.3.** *The Hessian $\nabla^2 E(\mathbf{x})$ satisfies the following $\forall \mathbf{x} \in \mathbb{R}^D$: it is positive definite; its largest eigenvalue and condition number are upper bounded by $\lambda + \|\bar{\mathbf{A}}_k\|^2$ and $1 + \|\bar{\mathbf{A}}_k\|^2/\lambda$, respectively; all its $D$ eigenvalues are greater or equal than $\lambda$; if $K \leq D$ then at least $D - K + 1$ eigenvalues equal $\lambda$.*

*Proof.* $\nabla^2 E(\mathbf{x})$ is symmetric positive definite since the softmax Hessian is positive semidefinite (from theorem 3.1) and the $\lambda$-term Hessian is positive definite. Hence, the eigenvalues of $\nabla^2 E(\mathbf{x})$ equal its singular values and have the form $\lambda + \lambda_d$, where $0 \leq \lambda_D \leq \cdots \leq \lambda_1$ are the eigenvalues of the softmax Hessian.

We can find an upper bound for $\lambda_1$ as follows. Let $\mathbf{u}$ be an eigenvalue of $\bar{\mathbf{A}}_k^T(\operatorname{diag}(\mathbf{p}) - \mathbf{pp}^T)\bar{\mathbf{A}}_k$, then:

$$\mathbf{u}^T \bar{\mathbf{A}}_k^T (\operatorname{diag}(\mathbf{p}) - \mathbf{pp}^T)\bar{\mathbf{A}}_k\mathbf{u}$$
$$= \mathbf{u}^T \bar{\mathbf{A}}_k^T \operatorname{diag}(\mathbf{p}) \bar{\mathbf{A}}_k\mathbf{u} - \mathbf{u}^T \bar{\mathbf{A}}_k^T \mathbf{pp}^T \bar{\mathbf{A}}_k\mathbf{u}$$
$$= \mathbf{u}^T \bar{\mathbf{A}}_k^T \operatorname{diag}(\mathbf{p}) \bar{\mathbf{A}}_k\mathbf{u} - \|\mathbf{p}^T \bar{\mathbf{A}}_k\mathbf{u}\|^2$$
$$\leq \mathbf{u}^T \bar{\mathbf{A}}_k^T \operatorname{diag}(\mathbf{p}) \bar{\mathbf{A}}_k\mathbf{u}$$

so its largest eigenvalue $\lambda_1$ is smaller or equal than the largest eigenvalue of $\bar{\mathbf{A}}_k^T \operatorname{diag}(\mathbf{p}) \bar{\mathbf{A}}_k$, which equals $\|\bar{\mathbf{A}}_k^T \operatorname{diag}(\mathbf{p}) \bar{\mathbf{A}}_k\|$. But $\|\bar{\mathbf{A}}_k^T \operatorname{diag}(\mathbf{p}) \bar{\mathbf{A}}_k\| \leq \|\bar{\mathbf{A}}_k\|^2\|\operatorname{diag}(\mathbf{p})\| < \|\bar{\mathbf{A}}_k\|^2$ (since $\mathbf{p}$ is in the interior of the regular simplex, so its largest element is smaller than 1). Consequently $\operatorname{cond}(\nabla^2 E(\mathbf{x})) = (\lambda + \lambda_1)/(\lambda + \lambda_D) \leq (\lambda + \lambda_1)/\lambda \leq 1 + \|\bar{\mathbf{A}}_k\|^2/\lambda$. Note that $\|\bar{\mathbf{A}}_k\|^2$ equals the largest eigenvalue of $\bar{\mathbf{A}}_k\bar{\mathbf{A}}_k^T$ (or $\bar{\mathbf{A}}_k^T\bar{\mathbf{A}}_k$).

If $K \leq D$ then $\operatorname{rank}(\bar{\mathbf{A}}_k) < K$, since $\bar{\mathbf{A}}_k$ contains one row of zeros, and the rank of the softmax Hessian (which involves a product with $\bar{\mathbf{A}}_k$) is at most $K - 1$, so $0 = \lambda_D = \cdots = \lambda_K \leq \lambda_{K-1} \leq \cdots \leq \lambda_1$. Hence $\nabla^2 E(\mathbf{x})$ has at least $D - K + 1$ eigenvalues equal to $\lambda$ and the rest are all greater or equal than $\lambda$. □

**Theorem 3.4.** *The gradient $\nabla E(\mathbf{x})$ is Lipschitz continuous with Lipschitz constant $L = \lambda + \|\bar{\mathbf{A}}_k\|^2$.*

*Proof.* This follows from the fact that, since $E(\mathbf{x})$ is convex and twice continuously differentiable, then $L$ is a Lipschitz constant of $\nabla E(\mathbf{x})$ iff $\|\nabla^2 E(\mathbf{x})\| \leq L \ \forall \mathbf{x} \in \mathbb{R}^D$ (which itself is a direct consequence of the mean value theorem). From theorem 3.3, $\|\nabla^2 E(\mathbf{x})\| \leq \lambda + \|\bar{\mathbf{A}}_k\|^2$, which we can take as $L$. □

**Theorem 3.5.** *If $\lambda \gg 1$ then $\nabla^2 E(\mathbf{x}) \approx \lambda\mathbf{I} \ \forall \mathbf{x} \in \mathbb{R}^D$. If $\lambda \ll 1$ then $\nabla^2 E(\mathbf{x}^*) \approx \lambda\mathbf{I}$.*

*Proof.* For $\lambda \gg 1$ the result follows because the softmax Hessian is bounded (from theorem 3.3). For $\lambda \ll 1$, the softmax probabilities satisfy $p_k(\mathbf{x}^*) \approx 1$ and $p_j(\mathbf{x}^*) \approx 0$ if $j \neq k$, so $\operatorname{diag}(\mathbf{p}(\mathbf{x}^*)) - \mathbf{p}(\mathbf{x}^*)\mathbf{p}(\mathbf{x}^*)^T \approx \mathbf{0}$ and the result follows. □

**Theorem 3.6.** *The inverse Hessian can be written as:*

$$\left(\nabla^2 E\right)^{-1} = \left(\lambda\mathbf{I} + \bar{\mathbf{A}}_k^T(\operatorname{diag}(\mathbf{p}) - \mathbf{pp}^T)\bar{\mathbf{A}}_k\right)^{-1} = \mathbf{H}^{-1} + \frac{(\mathbf{H}^{-1}\mathbf{v})(\mathbf{H}^{-1}\mathbf{v})^T}{1 - \mathbf{v}^T\mathbf{H}^{-1}\mathbf{v}} \tag{5}$$

*where $\mathbf{v} = \bar{\mathbf{A}}_k^T\mathbf{p}$ and $\mathbf{H}^{-1} = \frac{1}{\lambda}\left(\mathbf{I} - \bar{\mathbf{A}}_k^T\left(\bar{\mathbf{A}}_k\bar{\mathbf{A}}_k^T + \lambda\operatorname{diag}(\mathbf{p})^{-1}\right)^{-1}\bar{\mathbf{A}}_k\right)$. Likewise:*

$$-\left(\nabla^2 E\right)^{-1}\nabla E = -\lambda\left(\nabla^2 E\right)^{-1}(\mathbf{x} - \overline{\mathbf{x}}) - \frac{1}{1 - \mathbf{v}^T\mathbf{H}^{-1}\mathbf{v}}\mathbf{H}^{-1}\mathbf{v}. \tag{6}$$

*Proof.* Calling $\mathbf{v} = \bar{\mathbf{A}}_k^T \mathbf{p}$ and $\mathbf{H} = \lambda \mathbf{I} + \bar{\mathbf{A}}_k^T \operatorname{diag}(\mathbf{p}) \bar{\mathbf{A}}_k$, we can write the gradient as $\nabla E = \lambda(\mathbf{x} - \overline{\mathbf{x}}) + \mathbf{v}$ and the Hessian as $\nabla^2 E = \mathbf{H} - \mathbf{v}\mathbf{v}^T$. The results follow from applying the Sherman-Morrison-Woodbury formula $(\mathbf{A} + \mathbf{BCD})^{-1} = \mathbf{A}^{-1} - \mathbf{A}^{-1}\mathbf{B}(\mathbf{C}^{-1} + \mathbf{DA}^{-1}\mathbf{B})^{-1}\mathbf{DA}^{-1}$ twice: first to $\nabla^2 E$ by taking $\mathbf{A} = \mathbf{H}$, $\mathbf{B} = \mathbf{D}^T = \mathbf{v}$ and $\mathbf{C} = -1$, and then to $\mathbf{H}$ by taking $\mathbf{A} = \lambda \mathbf{I}$, $\mathbf{B}^T = \mathbf{D} = \bar{\mathbf{A}}_k$ and $\mathbf{C} = \operatorname{diag}(\mathbf{p})$. $\qquad\square$

Although the expressions above look complicated, they reduce the computation of the Newton direction by requiring a $K \times K$ inverse (of $\bar{\mathbf{A}}_k \bar{\mathbf{A}}_k^T + \lambda \operatorname{diag}(\mathbf{p})^{-1}$), rather than a $D \times D$ one (of $\nabla^2 E$). Since typically $K \ll D$, this is an enormous savings. Unfortunately, it is not possible to speed this up even further and avoid inverses altogether by caching a factorization of $\bar{\mathbf{A}}_k \bar{\mathbf{A}}_k^T + \lambda \operatorname{diag}(\mathbf{p})^{-1}$, because it is a diagonal update. Note that we never actually invert any matrix: for reasons of numerical stability and efficiency, expressions of the form $\mathbf{A}^{-1}\mathbf{b}$ are computed by solving a linear system $\mathbf{Ax} = \mathbf{b}$ rather than by explicitly computing $\mathbf{A}^{-1}$ and multiplying it times $\mathbf{b}$ (in Matlab, `x = A\b` rather than `x = inv(A)*b`).

Having obtained the solution $\mathbf{x}^*(\lambda)$ for a value of $\lambda$, we obtain the corresponding softmax probabilities $p_i(\mathbf{x}^*(\lambda))$ for $i = 1, \ldots, K$ by evaluating the softmax function at $\mathbf{x}^*$. Also, from the definition of the objective function in eq. (2), we have that $p_i(\mathbf{x}^*(\lambda)) = \exp\left(\frac{\lambda}{2}\|\mathbf{x}^*(\lambda) - \overline{\mathbf{x}}\|^2 - E(\mathbf{x}^*(\lambda))\right)$.

### 3.3 Convergence and rate of convergence

Theorem 3.8 states that Newton's method with a line search (l.s.) will converge to the minimizer of $E$ from any starting point (global convergence). It is a consequence of our results above and the following theorem[3].

**Theorem 3.7.** *Consider a function $f\colon \mathbb{R}^D \to \mathbb{R}$, not necessarily convex, bounded below and continuously differentiable in $\mathbb{R}^D$, and with $L$-Lipschitz continuous gradient (i.e., $\|\nabla f(\mathbf{x}) - \nabla f(\mathbf{y})\| \le L\|\mathbf{x} - \mathbf{y}\| \ \forall \mathbf{x}, \mathbf{y} \in \mathbb{R}^D$ and $L > 0$). Consider an iteration of the form $\mathbf{x}_{k+1} = \mathbf{x}_k + \alpha_k \mathbf{s}_k$, where $\mathbf{x}_0 \in \mathbb{R}^D$ is any starting point, $\mathbf{s}_k$ is a descent direction (i.e., $\mathbf{s}_k^T \nabla f(\mathbf{x}_k) < 0$ if $\nabla f(\mathbf{x}_k) \ne \mathbf{0}$) and the step size $\alpha_k$ satisfies the Wolfe conditions. Then $\sum_{k \ge 0} \cos^2 \theta_k \|\nabla f(\mathbf{x}_k)\|^2 < \infty$, where $\cos \theta_k = -\mathbf{s}_k^T \nabla f(\mathbf{x}_k) / \|\mathbf{s}_k\|\|\nabla f(\mathbf{x}_k)\|$.*
*Now assume $\mathbf{s}_k = -\mathbf{B}_k^{-1} \nabla f(\mathbf{x}_k)$ where, for each $k = 0, 1, 2 \ldots$, $\mathbf{B}_k$ is positive definite and $\operatorname{cond}(\mathbf{B}_k) \le M$ for some $M > 0$. Then $\|\nabla f(\mathbf{x}_k)\| \to \mathbf{0}$ as $k \to \infty$.*

*Proof.* This theorem is an amalgamation of theorem 3.2 (Zoutendijk's theorem) and exercise 3.5 in Nocedal & Wright (2006). The bound $\operatorname{cond}(\mathbf{B}_k) \le M$ implies a bound $\cos \theta_k \ge \frac{1}{M} > 0$, from which the second result follows. The theorem also holds for other l.s. conditions such as the strong Wolfe conditions or the Goldstein conditions. $\qquad\square$

**Theorem 3.8.** *Consider the objective function $E(\mathbf{x})$ of eq. (2) and an iteration of the form $\mathbf{x}_{k+1} = \mathbf{x}_k - \alpha_k \left(\nabla^2 E(\mathbf{x}_k)\right)^{-1} \nabla E(\mathbf{x}_k)$ for $k = 0, 1, 2 \ldots$, where $\mathbf{x}_0 \in \mathbb{R}^D$ is any starting point and the step size $\alpha_k$ satisfies the Wolfe conditions. Then $\lim_{k \to \infty} \mathbf{x}_k = \mathbf{x}^*$, the unique minimizer of $E$, and $\lim_{k \to \infty} \|\nabla f(\mathbf{x}_k)\| = \mathbf{0}$.*

*Proof.* Since $\nabla^2 E(\mathbf{x}_k)$ is positive definite, $-\left(\nabla^2 E(\mathbf{x}_k)\right)^{-1} \nabla E(\mathbf{x}_k)$ is a descent direction. The function $E(\mathbf{x})$ is nonnegative (hence lower bounded) and continuously differentiable in $\mathbb{R}^D$. Its gradient $\nabla E(\mathbf{x})$ is Lipschitz continuous in $\mathbb{R}^D$ (from theorem 3.4) and its Hessian's condition number is upper bounded (from theorem 3.3). Hence, from theorem 3.7, $\|\nabla f(\mathbf{x}_k)\| \to \mathbf{0}$ as $k \to \infty$. Since $E$ is strongly convex and differentiable, there can only be one point with zero gradient, which is the global minimizer $\mathbf{x}^*$, hence $\mathbf{x}_k \to \mathbf{x}^*$. $\qquad\square$

Further, theorem 3.5 in Nocedal & Wright (2006) states that, if the starting point $\mathbf{x}_0$ is sufficiently close to $\mathbf{x}^*$ (which can always be achieved by the l.s.) and if using unit step sizes, then both the sequences of iterates $\{\mathbf{x}_k\}$ and of gradients $\{\nabla E(\mathbf{x}_k)\}$ converge *quadratically* to $\mathbf{x}^*$ and $\mathbf{0}$, respectively. Although the theorem we give requires a l.s. with Wolfe conditions, a backtracking search that always tries the unit step size first also works, because it is sufficient to take us near enough the minimizer and then using unit step sizes converges quadratically.

---

[3]In this section (and elsewhere in the paper when referring to an iterate), we use the subindex $k$ to indicate the $k$th iterate in the optimization algorithm. This should not be confused with the use of $k$ to indicate a class index, as in eq. (2).

For gradient descent, global convergence is also assured with a l.s. (by taking $\mathbf{B}_k = \mathbf{I}$ in theorem 3.7) or with a fixed step size of $1/L$ (Nesterov, 2004). For other algorithms (Polak-Ribière CG, and (L-)BFGS) it is harder to give robust global convergence results (Nocedal & Wright, 2006), but in any case they are inferior to Newton's method in our case.

### 3.4 Computational complexity

For Newton's method, the dominant costs are as follows. There is a setup cost of $\mathcal{O}(DK^2)$ to compute $\bar{\mathbf{A}}_k \bar{\mathbf{A}}_k^T$. Per iteration, we have a cost of $\mathcal{O}(D(2K + 1))$ for the gradient, $\mathcal{O}(2K^3 + 6KD)$ for the Newton direction, and $\mathcal{O}(D(K + 1))$ per backtracking step; our experiments show that most iterations use a single backtracking step (see fig. 3). If $D \gg K$ one Newton iteration costs about 3 gradient computations.

### 3.5 Solving for a path over $\lambda$

It is of interest to solve problem (2) not just for a single value of $\lambda$, but for a range of values of $\lambda$. This can provide a set of counterfactual explanations at different distances and with different target class probabilities, which a user may examine and choose from. It is also a way to solve the constrained form of the problem, namely "$\min_{\mathbf{x}} \|\mathbf{x} - \overline{\mathbf{x}}\|$ s.t. $g_k(\mathbf{x}) \leq \alpha$", since tracing this over $\alpha$ corresponds to tracing (2) over $\lambda$.

Using our algorithm to optimize for each $\lambda$ value, solving for a path of $\lambda$ from large to small (say $10^3$ to $10^{-5}$) is very efficient by using warm-start over $\lambda$. The first $\lambda$ value is initialized from $\mathbf{x} = \overline{\mathbf{x}}$, to which the solution is very close. If $\lambda$ changes slowly, i.e., we follow a fine path, a step size of 1 in Newton's method typically works, but we try more $\lambda$ values. Alternatively, we can change $\lambda$ more quickly, i.e., we follow a coarse path and try few $\lambda$ values, but we will need more Newton iterations per $\lambda$. Either way, since the algorithm is very fast, this does not matter much, and solving for the whole path is just a bit slower than solving for a single $\lambda$. We illustrate this in our experiments (see fig. 4).

### 3.6 Discussion

An inverse classification problem (e.g. a counterfactual explanation) requires a definition of distance or norm in instance space that measures the cost of changing the source instance. In general, the norm depends on the application. For example, some works in the adversarial examples literature have used the $\ell_2$ norm, such as Szegedy et al. (2014, section 4.1) or Moosavi-Dezfooli et al. (2016, section 1). Other works have used the $\ell_1$ or $\ell_\infty$ norm. In this work we use only the $\ell_2$ norm because, together with the logistic regression or softmax classifier, it makes a very efficient computation possible for its unique solution. We are not aware of a similarly efficient solution for other norms. This is particularly useful in real-time or interactive scenarios where fast feedback matters. For instance, in a credit approval or document moderation system, a user may want to understand which minimal changes to the input, such as adjusting an income-related feature or modifying a small portion of the text, would be sufficient to change the model's decision. The efficiency of our approach makes it possible to compute such counterfactual explanations on demand, allowing users or practitioners to explore different "what-if" scenarios interactively.

Although the objective function $E$ is nonlinear, the special structure of its Hessian makes minimizing it very effective. Firstly, since $E$ is strongly convex, there is a unique solution no matter the value of $\lambda$ or the target class $k$. Second, if $K < D$ (which is the typical case in practice, fewer classes than features) then the Hessian has $D - K + 1$ eigenvalues equal to $\lambda$ and $K - 1$ greater or equal than $\lambda$. This means that the Hessian is much better conditioned than it would otherwise be; it is "round" in all but $K - 1$ directions, and all optimization methods will benefit from that. That said, using the curvature information is still important to take long steps. Finally, the Hessian is actually round everywhere if $\lambda \gg 1$ or near the minimizer if $\lambda \ll 1$.

This special structure also makes Newton's method unusually convenient. In practice, Newton's method is rarely applicable in direct form for two important reasons: 1) the Hessian is often not positive definite and hence can create directions that are not descent, so it needs modification, which is computationally costly (e.g. it may require factorizing it). 2) Computing the Newton direction requires computing the Hessian (in $\mathcal{O}(D^2)$ time and memory) and solving a linear system based on it (in $\mathcal{O}(D^3)$ time). To make this feasible with large $D$ requires approximating Newton's method, which leads to taking worse steps and losing its quadratic

convergence properties. None of this is a problem here: our Hessian is positive definite everywhere, and computing the Newton direction (via theorem 3.6) requires a linear system of a $K \times K$ matrix, where $K$ is in practice small (less than 100 is the vast majority of applications), without ever forming a $D \times D$ matrix, which then scales to very large $D$. Finally, while Newton's method still requires a line search for the step size to ensure the iterates descend, near the minimizer a unit step works and leads to quadratic convergence. As a result we achieve the best of all possible worlds: global convergence (from any starting point); very fast iterations, scalable to high feature dimensionalities and many classes; and quadratic convergence order, which means it is possible to reach near machine-precision accuracy.

This theoretical argument strongly suggests that no other (first-order) method can compete with Newton's method in this case, particularly if we seek a highly accurate solution (unless we use a huge number of classes $K$), and this is clearly seen in our experiments. The runtime ranges from milliseconds to a second over problems with feature dimension ranging from $10^3$ to $10^5$ and tens of classes.

This may surprise readers used to optimizing convex problems in machine learning using gradient descent with a fixed step size and no line search (which converges linearly with strongly convex problems), or even an online form of gradient descent. This is convenient when the objective function is costly to evaluate, for example consisting of a sum over a large training set. But the setting here is different: it is as if we had a single training instance. Hence, using a quadratically convergent method with a line search is far faster.

## 4 Inverse classification as optimization: logistic regression

Logistic regression corresponds to the case $K = 2$ of the softmax classifier. All the previous results apply, but in this case we can find $\mathbf{x}^*$ in closed form[4], as given in theorem 4.2. Let us take $k = 1$ as the target class w.l.o.g. and define $\mathbf{w} = \mathbf{a}_2 - \mathbf{a}_1$ and $w_0 = b_2 - b_1$ given the softmax parameters $\{\mathbf{A}, \mathbf{b}\}$, or we can just think of $\{\mathbf{w}, w_0\}$ as the weight vector and bias of the logistic regression model, where $\mathbf{w}$ points towards class 1. Then we have

$$p_1(\mathbf{x}) = \frac{1}{1 + e^{\mathbf{w}^T \mathbf{x} + w_0}} \in (0, 1) \tag{7a}$$

$$p_2(\mathbf{x}) = 1 - p_1(\mathbf{x}) \tag{7b}$$

$$g_1(\mathbf{x}) = \log\left(1 + e^{\mathbf{w}^T \mathbf{x} + w_0}\right) \tag{7c}$$

$$\nabla_{\mathbf{x}} E(\mathbf{x}; \lambda) = \lambda(\mathbf{x} - \overline{\mathbf{x}}) + (1 - p_1(\mathbf{x}))\mathbf{w} \tag{7d}$$

$$\nabla_{\mathbf{xx}}^2 E(\mathbf{x}; \lambda) = \lambda \mathbf{I} + p_1(\mathbf{x})(1 - p_1(\mathbf{x}))\mathbf{w}\mathbf{w}^T. \tag{7e}$$

Define the function $\phi(\alpha, \beta)$: $\mathbb{R}_+ \times \mathbb{R} \to (0, 1)$ as the unique root of the equation $t = 1/(1 + \exp(\alpha t + \beta))$ for $\alpha > 0$ and $\beta \in \mathbb{R}$. The next theorem shows $\phi$ is well defined[5].

**Theorem 4.1.** *Let $\alpha > 0$, $\beta \in \mathbb{R}$ and $h(t) = t - 1/(1 + \exp(\alpha t + \beta))$ for $t \in \mathbb{R}$. Then $h$ has a unique root $t^*$, $h(t^*) = 0$, and $t^* \in (0, 1)$.*

*Proof.* This holds because $h'(t) > 0 \; \forall t \in \mathbb{R}$, so $h$ is strictly monotonically increasing, and $h(0) < 0$ and $h(1) > 0$. □

**Theorem 4.2.** *The unique minimizer of $E(\mathbf{x}; \lambda)$ (from eq. (2)) is $\mathbf{x}^* = \overline{\mathbf{x}} - \frac{1}{\lambda}(1 - p_1^*)\mathbf{w}$ where $p_1^* = \phi(\alpha, \beta)$, $\alpha = \frac{1}{\lambda}\|\mathbf{w}\|^2 > 0$ and $\beta = \mathbf{w}^T \overline{\mathbf{x}} + w_0 - \alpha \in \mathbb{R}$.*

*Proof.* Since $E$ is strongly convex and continuously differentiable, $\mathbf{x}^* \in \mathbb{R}^D$ is the unique solution of the nonlinear system of $D$ equations $\nabla E(\mathbf{x}^*) = \lambda(\mathbf{x}^* - \overline{\mathbf{x}}) + (1 - p_1(\mathbf{x}^*))\mathbf{w} = \mathbf{0} \Leftrightarrow \mathbf{x}^* = \overline{\mathbf{x}} - \frac{1}{\lambda}(1 - p_1(\mathbf{x}^*))\mathbf{w}$. Hence $\mathbf{w}^T \mathbf{x}^* + w_0 = \mathbf{w}^T \overline{\mathbf{x}} + w_0 - \frac{1}{\lambda}(1 - p_1(\mathbf{x}^*))\|\mathbf{w}\|^2$. Since, from eq. (7), $p_1(\mathbf{x}) = 1/(1 + \exp(\mathbf{w}^T \mathbf{x} + w_0))$,

---

[4]By "closed form" we mean in terms of the scalar function $\phi(\alpha, \beta)$ (even though computing the latter requires an iterative algorithm).

[5]The function $\phi(\alpha, \beta)$ for $\alpha < 0$ was studied in (Carreira-Perpiñán & Williams, 2003, pp. 5–6), where it arises in the context of finding the maximum of a Gaussian mixture with two components. In that case there can be one, two or three roots depending on $\alpha$ and $\beta$, so $\phi$ can be multivalued, unlike in our case, where it is univalued.

substituting in it the previous expression yields $p_1^* = 1/(1 + \exp{(\alpha p_1^* + \beta)})$. This is a scalar equation for $p_1^* \in (0, 1)$ whose solution, from theorem 4.1, is $p_1^* = \phi(\alpha, \beta)$. □

Theorem 4.2 shows that the problem of minimizing $E$ in $D$ dimensions *is really a problem in one dimension*, and the solution $\mathbf{x}^*$ lies along the ray emanating from $\overline{\mathbf{x}}$ along $\mathbf{w}$. Hence, to find the minimizer of eq. (2), we compute $\alpha$, $\beta$, $p_1^*$ and $\mathbf{x}^*$ using theorem 4.2. To compute $p_1^* = \phi(\alpha, \beta)$, we have to solve a scalar equation. This can be done using a robustified version of Newton's method in one variable (e.g. by falling back to a bisection step if the Newton step jumps out of the current root bracket, as in Vladymyrov & Carreira-Perpiñán (2013)). Convergence to machine precision occurs in a few iterations, in $\mathcal{O}(1)$. The overall computational complexity is $\mathcal{O}(D)$, dominated by reading the data ($\overline{\mathbf{x}}$, $\mathbf{w}$) and doing two vector-vector products ($\mathbf{w}^T\overline{\mathbf{x}}$ and $\mathbf{w}^T\mathbf{w}$). For $D$ ranging from $10^3$ to $10^5$, the runtime ranges from microseconds to milliseconds. As shown in our experiments, this is far faster than running Newton's method in the $D$-dimensional space.

## 5 Experiments

Our experiments are very straightforward and agree well with the theory. In short, Newton's method consistently achieves the fastest runtime of any algorithm, by far, while reaching the highest precision possible. For logistic regression, the closed-form solution is even better. The appendix gives gives more details.

### 5.1 $K$-class softmax

We solve problems of the form (2) on 4 datasets of different characteristics: MNIST ($D = 784$, $K = 10$), of handwritten digit images with grayscale pixel features (LeCun et al., 1998); RCV1 ($D = 47\,236$, $K = 51$), of documents with TFIDF features (Lewis et al., 2004); VGGFeat64 ($D = 8\,192$, $K = 16$), of neural net features from the last convolutional layer of a pre-trained VGG16 deep net (Simonyan & Zisserman, 2015) on a subset of 16 classes of the ImageNet dataset (Deng et al., 2009), where each image has been resized to $64{\times}64$ pixels; and VGGFeat256 ($D = 131\,072$, $K = 16$), like VGGFeat64 but using images of $256{\times}256$ to create the features. For each dataset we trained a softmax classifier using `scikit-learn` (Pedregosa et al., 2011).

We evaluated several well-known optimization methods, and for each we use the line search that we found worked best. For Newton's method, BFGS and L-BFGS we use a backtracking line search with initial step 1 and backtracking factor $\rho = 0.8$. For gradient descent (GD) and Polak-Ribière conjugate gradient (CG), we use a more sophisticated line search that allows steps longer than 1 and ensures the Wolfe conditions hold (algorithm 3.5 in Nocedal & Wright (2006)). All methods were implemented by us in Matlab except for CG, which uses Carl Rasmussen's `minimize.m` very efficient implementation. For L-BFGS, we tried several values of its queue size and found $m = 4$ worked best in our datasets. Each method iterates until $\|\nabla E(\mathbf{x}_k)\| < 10^{-8}$, up to a maximum of $1\,000$ iterations (which only GD reaches, occasionally). The initial iterate was always the source instance $\mathbf{x}_0 = \overline{\mathbf{x}}$.

For each dataset, we generated 50 instances of problem (2), each given by a source instance $\overline{\mathbf{x}}$, a target class $k$ and a value of $\lambda$ (see details in the appendix). Fig. 1 shows a scatterplot of objective function value $E(\mathbf{x}^*)$, number of iterations and runtime in seconds for each method on each problem instance. It is clear that Newton's method works better than any other method, by far (note the plots are in log scale). It takes around 10 iterations to converge and between 1 and 100 ms runtime—a remarkable feat given that the problems are nonlinear and reach over $10^5$ variables. Next best are CG and L-BFGS, which take about 10 times longer; and finally, GD, which is far slower. BFGS is only applicable in small- to medium-size problems because it stores a Hessian matrix approximation explicitly; in our problems this makes it very slow even if it does not require many iterations.

Fig. 2 shows the learning curves for one specific problem instance per dataset. Most informative are those for $E(\mathbf{x}_k) - E(\mathbf{x}^*)$ and $\|\nabla E(\mathbf{x}_k)\|$, which clearly show the asymptotic convergence rate of each method (note that while $E$ must decrease monotonically at each iteration, the gradient norm need not). For linearly convergent methods (all except Newton's) the iterates in a log-log plot should trace a straight line of negative

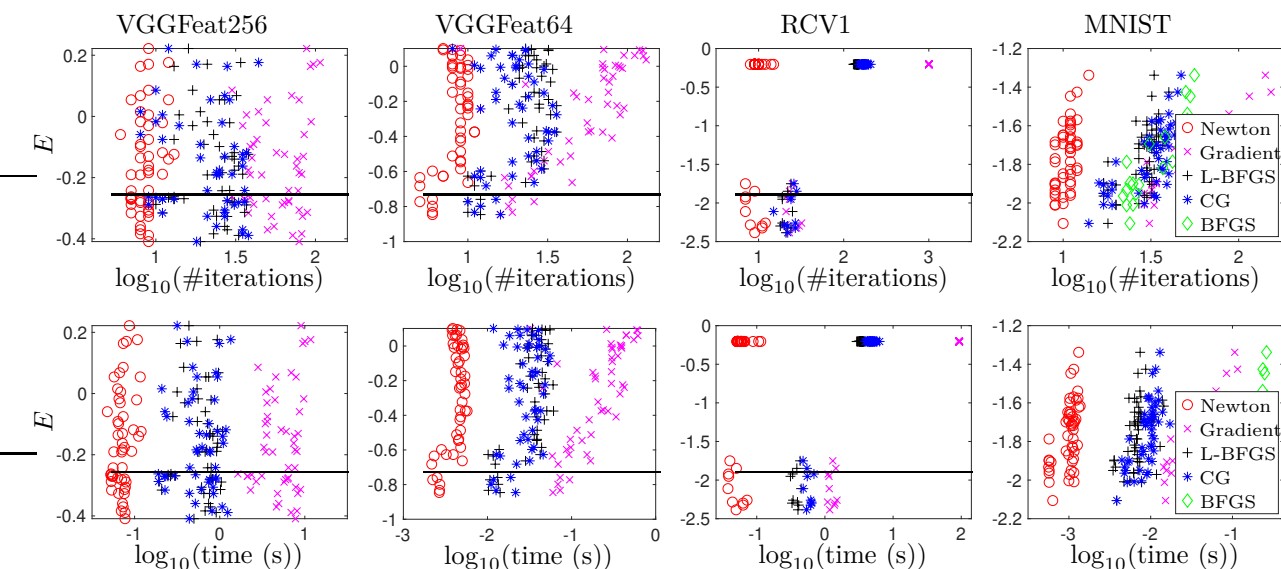

Figure 1: Performance of different optimization methods on a collection of 50 problem instances per dataset, in number of iterations (top) and runtime in seconds (below).

slope equal to the logarithm of the rate; for quadratically convergent methods (Newton's), they trace an exponential curve instead. This is clearly visible in the ends of the curves. The last few iterates of Newton's method double the number of correct decimal digits at each iteration, and quickly reach the precision limit of the machine.

Figures 3–4 further investigate Newton's method. The histogram in fig. 3 shows that the initial step (equal to 1) is accepted most times, and rarely are more than 5 trials needed. This confirms that a better line search that ensures the Wolfe conditions hold is unlikely to help much. Figure 4 (left plot) uses the number of iterations as a proxy for "difficulty" of the problem, by solving each problem for 100 values of $\lambda$. As predicted by theorem 3.5, problems with $\lambda \gg 1$ are easiest, followed by problems with $\lambda \ll 1$. Figure 4 (right table) compares, when solving a problem over a range of decreasing $\lambda$ values, whether to warm-start (i.e., initialize the next problem from the solution of the previous one) or not (i.e., initialize always from $\overline{\mathbf{x}}$). Warm-start is about $5\times$ faster and very few iterations are required for each $\lambda$ value. This makes it attractive to solve for a range of $\lambda$s and present the user an array of solutions of increasing target class probability $p_k(\mathbf{x}^*(\lambda))$.

## 5.2 Logistic regression

We created binary classification versions of the datasets used for the softmax (same training and test instances) by partitioning the classes into two groups and training a logistic regression to learn that. As expected, the closed-form solution was far faster (about $100\times$) than Newton's method.

## 6 Conclusion

Inverse classification problems such as counterfactual explanations and adversarial examples can be formulated as optimization problems. We have theoretically characterized a particular formulation (using the squared Euclidean distance) for two fundamental classifiers: logistic regression and softmax classifiers. The former admits a closed-form solution, while the latter can be solved extremely efficiently with Newton's method using a suitably reformulated Hessian. The fact that we can solve this so efficiently is due to the special form of the logistic regression and softmax classifiers, and to the formulation we use of the optimization problem. While other formulations are possible, their solution would not be as efficient as here. In our formulation, what is a $D$-dimensional optimization problem (where $D$ is the number of features)

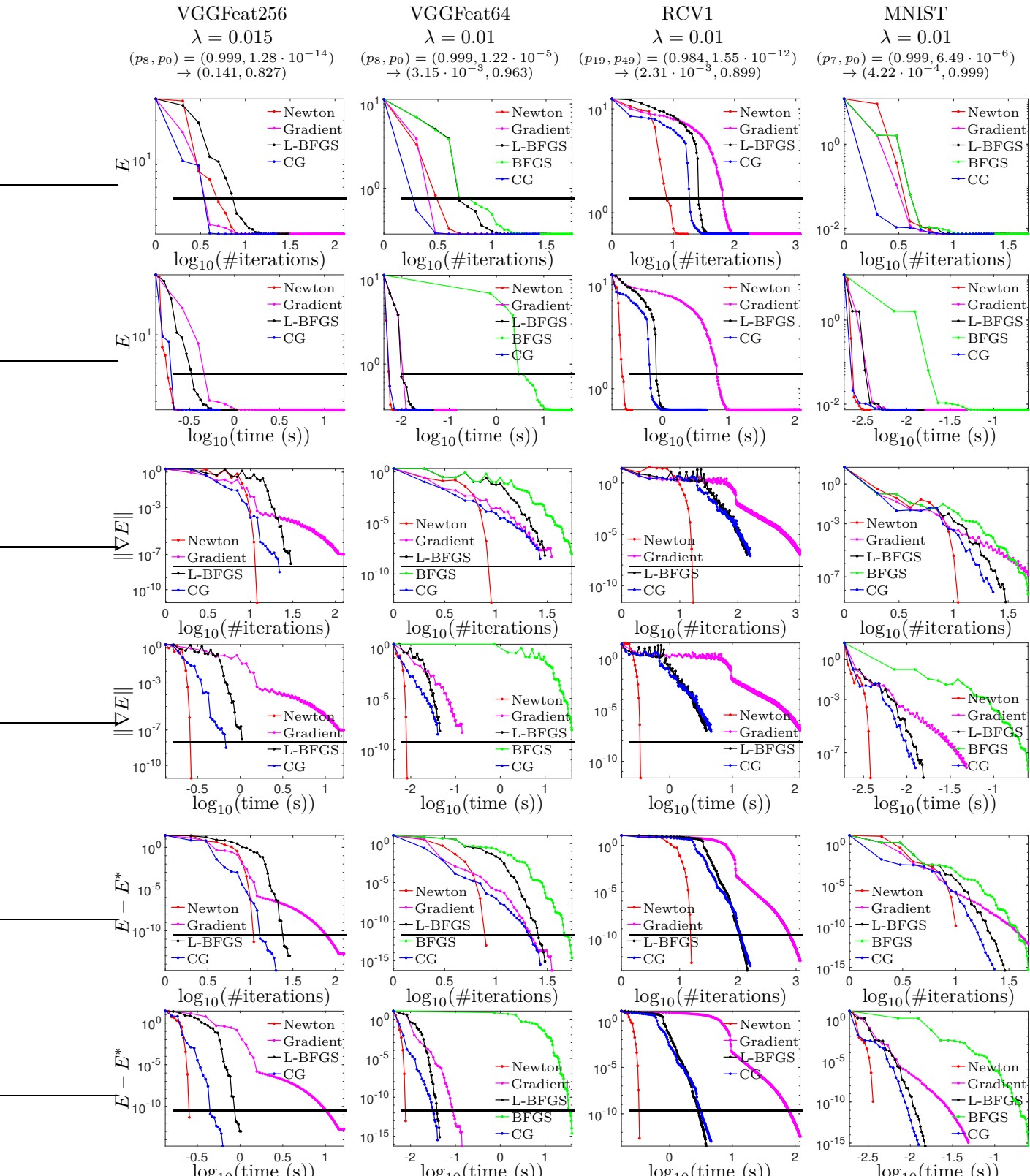

Figure 2: Learning curves for one problem instance per dataset (columns), indicated by its $\lambda$ value and class probabilities, encoded as $(p_{\overline{k}}, p_k) = (\text{at } \overline{\mathbf{x}}) \rightarrow (\text{at } \mathbf{x}^*)$. Rows 1–2, 3–4 and 5–6 show $E(\mathbf{x}_k)$, $\|\nabla E(\mathbf{x}_k)\|$ and $E(\mathbf{x}_k) - E(\mathbf{x}^*)$, respectively (where $\mathbf{x}^*$ is estimated by the last iterate of Newton's method). Within each pair of rows, we show number of iterations (above) and runtime in seconds (below). All plots are log-log.

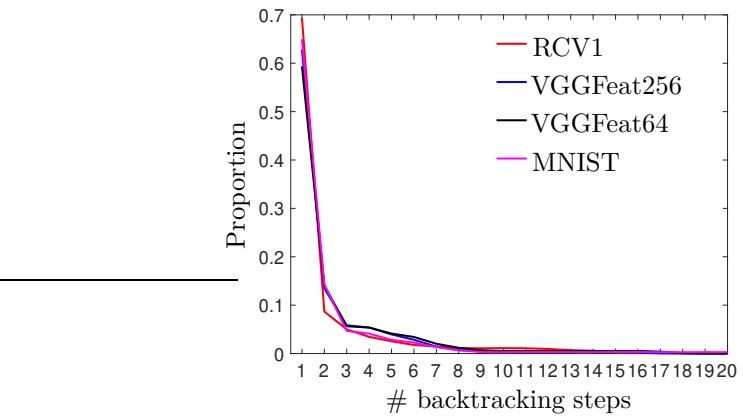

Figure 3: Histogram of the number of steps tried in the backtracking line search for Newton's method, for instances of different datasets.

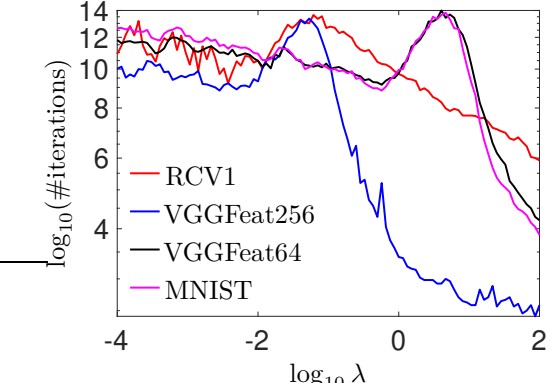

| Dataset | runtime (seconds) | |
| --- | --- | --- |
| | warm start | no warm start |
| MNIST | $0.021 \pm 0.002$ | $0.093 \pm 0.015$ |
| VGGFeat264 | $1.460 \pm 0.022$ | $6.147 \pm 0.791$ |
| VGGFeat64 | $0.079 \pm 0.004$ | $0.296 \pm 0.014$ |
| RCV1 | $1.351 \pm 0.031$ | $6.749 \pm 1.782$ |

Figure 4: *Left*: number of iterations for Newton's method as a function of $\lambda$ (averaged, for each $\lambda$, over the 50 problem instances of fig. 1). *Right*: total runtime (seconds) over the 100 values of $\lambda$ in the top plot with and without warm-start (average $\pm$ stdev over the 50 problems).

can essentially be reduced to a $K$-dimensional optimization problem (where $K$ is the number of classes) for the softmax classifier, and to a scalar problem for logistic regression. This is important because $K \ll D$ in most applications of interest. For both classifiers, we can solve the optimization to practically machine precision and scale to feature vectors of over $10^5$ dimensions and tens of classes with a runtime under one second. This makes it suitable for real-time or interactive applications. Future work includes handling a very large number of classes; constraints on the desired instance; and investigating other classifiers, distances and problem settings.

Code implementing the algorithms is available from the authors.

**Acknowledgments**

Work partially supported by NSF award IIS–2007147.

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

## A    Datasets and experimental setup

We use the following 4 datasets (summarized in table 1):

**MNIST** (LeCun et al., 1998) It consists of 28×28 grayscale images of handwritten digits of classes 0 to 9, represented as a feature vector of pixel values in [0,1] of dimension $D = 784$. There are 60 000 training instances and 10 000 test instances.

**RCV1** (Lewis et al., 2004) It consists of newswire stories manually categorized into $K = 51$ classes and made available by Reuters, Ltd. for research purposes. Each document is represented as a vector of TFIDF features of dimension $D = 47 236$. There are 15 564 training instances and 518 571 test instances.

**VGGFeat64** Each instance corresponds to an image of 64×64 pixels, categorized into $K = 16$ object classes, a subset of man-made and natural objects from the ImageNet dataset (Deng et al., 2009). Each image is represented by a feature vector of dimension $D = 8 192$, corresponding to the neural net features from the last convolutional layer of a pre-trained VGG16 deep net (Simonyan & Zisserman, 2015) (on the subset of 16 classes). There are 16 000 training instances and 3 200 test instances.

**VGGFeat256** Like VGGFeat64, but using images of 256×256 to create a vector of $D = 131 072$ features.

We created binary classification versions of the MNIST, RCV1 and VGGFeat256 datasets for use with logistic regression. They have the same training and test instances and feature vectors and only differ in the class labels, which are regrouped from $K$ into 2. For MNIST, we use even vs odd digits. For RCV1, we use classes 0–25 vs 26–51. For VGGFeat256, we use man-made objects (7 classes) vs natural objects (9 classes).

For each dataset we trained a softmax classifier using `scikit-learn` (Pedregosa et al., 2011). The training and test error are in table 1.

In some experiments (e.g. in fig. 1) we use a collection of 50 inverse classification problem instances per dataset, each consisting of a triplet $(\overline{\mathbf{x}}, k, \lambda)$ in eq. (2). We generated them at random with the goal of having some diversity in the problem difficulty, as follows. For 40 of the problems, we picked $\overline{\mathbf{x}}$ uniformly at random from the training set; we picked $k$ as the class having lowest softmax probability for $\overline{\mathbf{x}}$ (i.e., the "farthest" class, having a near-zero probability); and we set $\lambda = 0.01$ (which makes the minimizer achieve a softmax probability for class $k$ above 0.95 or so). For the remaining 10 problems, we picked $\overline{\mathbf{x}}$ uniformly at random from the training set; we picked $k$ as the class having the second largest softmax probability for $\overline{\mathbf{x}}$ (i.e., the "closest" class); and we set $\lambda = 0.1$ (which again makes the minimizer achieve a softmax probability for class $k$ above 0.95 or so). For VGGFeat256 we set instead $\lambda = 0.006$ and 0.08, respectively. In fig. 4, we used the 50 problems for each dataset but for each we kept the $(\overline{\mathbf{x}}, k)$ part and varied $\lambda$ in $[10^{-4}, 10^2]$.

All our experiments were ran using Matlab in a PC with an Intel Core i7 processor (8 cores) and 16 GB RAM.

Table 1: Datasets and softmax classifiers.

| Dataset | $D$ | $K$ | Training set $N$ | error (%) | Test set $N$ | error (%) |
|---|---|---|---|---|---|---|
| MNIST | 784 | 10 | 60 000 | 6.14 | 10 000 | 7.35 |
| RCV1 | 47 236 | 51 | 15 564 | 7.03 | 518 571 | 14.52 |
| VGGFeat64 | 8 192 | 16 | 16 000 | 0.00 | 3 200 | 6.54 |
| VGGFeat256 | 131 072 | 16 | 16 000 | 0.00 | 3 200 | 3.07 |
| MNIST ($K = 2$) | 784 | 2 | 60 000 | 15.60 | 10 000 | 15.83 |
| RCV1 ($K = 2$) | 47 236 | 2 | 15 564 | 4.23 | 518 571 | 5.16 |
| VGGFeat256 ($K = 2$) | 131 072 | 2 | 16 000 | 0.00 | 3 200 | 0.44 |

# B    Looking at the digits of accuracy in the solution

As discussed in the main text, Newton's method converges quadratically while all other methods converge linearly (or superlinearly for BFGS). This is evident in fig. 2, where the $E_k - E^*$ (or $\|\nabla E_k\|$) curve is an exponential curve for Newton's method but a straight line for linearly convergent methods. Table 2 shows this more dramatically by printing explicitly the value of $E_k - E^*$ so we can see the decimals of accuracy achieved. It is clear that, once near the minimizer, Newton's method enters an asymptotic regime where it converges quadratically, and the number of correct digits roughly *doubles* at each iteration, so we reach machine precision in just a few iterations. Linearly convergent methods require far many more iterations to achieve the same accuracy.

Table 2: Numerical values of $E(\mathbf{x}_k) - E(\mathbf{x}^*)$ for the iterates $k = 1, 2 \ldots$ for the RCV1 dataset example of fig. 2. Note that not all values of $k$ are shown. Blank elements mean the method already reached the stopping criterion.

| $k$ | Newton | Gradient descent | L-BFGS | Conjugate gradient |
|---|---|---|---|---|
| 1 | 11.8619041837028210 | 11.8619041837028210 | 11.8619041837028210 | 11.8619041837028210 |
| 2 | 9.9372386648974036 | 9.4302539222109605 | 11.5799873531340047 | 7.8983750132722008 |
| 3 | 9.0808867584076030 | 8.7110587612082764 | 10.7583569748493861 | 7.6657393532552422 |
| 4 | 8.8305942589710611 | 8.3881330123967608 | 9.9349688191421208 | 7.4857154746015588 |
| 5 | 6.0417478625925387 | 8.1562216346068226 | 9.4216784942161258 | 7.3181433942342551 |
| 6 | 2.6591874389232428 | 7.9932275656377580 | 8.8938728977689721 | 6.9227281220705681 |
| 7 | 1.2890680769334155 | 7.9712244558701926 | 8.8149865066619277 | 6.4034389561568101 |
| 8 | 0.6853217955710699 | 7.6718893687727281 | 8.2623248226038566 | 6.0762883180276770 |
| 9 | 0.5424719611123440 | 7.5005660590871734 | 8.0888003708399339 | 5.8814411631701109 |
| 10 | 0.0760466660936738 | 7.3211346697357822 | 7.8860714868005122 | 5.6633127311506914 |
| 11 | 0.0168636110015056 | 7.2410781355651777 | 7.4850916049060583 | 5.3105596069985168 |
| 12 | 0.0031375192229656 | 7.0977915521472728 | 7.2430847483604373 | 4.9945815453849001 |
| 13 | 0.0003850423026792 | 7.0388488056588061 | 6.7245677562655546 | 4.8097802381394512 |
| 14 | 0.0000147435815694 | 6.8631000669546767 | 6.6127580406687594 | 4.6308326737056191 |
| 15 | 0.0000000349853287 | 6.7057687379199944 | 6.4090847162496329 | 4.2268346415684688 |
| 16 | 0.0000000000002218 | 6.5381468797087248 | 6.0836575796218124 | 4.0361704289760212 |
| 17 | 0.0000000000000000 | 6.4878228940302725 | 5.5272242498443811 | 2.3805364947786689 |
| 25 |  | 5.3636636844729431 | 2.6898091938913939 | 0.0629701004269668 |
| 100 |  | 0.0003447036460872 | 0.0000000012869300 | 0.0000000009660868 |
| 145 |  | 0.0000432782256198 | 0.0000000000000328 | 0.0000000000010827 |
| 160 |  | 0.0000247039692411 |  | 0.0000000000001663 |
| 165 |  | 0.0000207129199613 |  |  |
| 500 |  | 0.0000000231646672 |  |  |
| 1000 |  | 0.0000000000059133 |  |  |
| 1174 |  | 0.000000000003579 |  |  |

## C   Type of line search

We consider 3 types of line search (l.s.) to determine the step size $\alpha_k$ given a search direction $\mathbf{s}_k \in \mathbb{R}^D$:

**Wolfe**  This is a l.s. that finds a step size that satisfies the strong Wolfe conditions:

- $E(\mathbf{x}_k + \alpha_k \mathbf{s}_k) \leq E(\mathbf{x}_k) + c_1 \alpha_k \nabla E(\mathbf{x}_k)^T \mathbf{s}_k$ ("sufficient decrease in objective function")
- $\left| \nabla E(\mathbf{x}_k + \alpha_k \mathbf{s}_k)^T \mathbf{s}_k \right| \leq c_2 \left| \nabla E(\mathbf{x}_k)^T \mathbf{s}_k \right|$ ("flatter gradient")

where $0 < c_1 < c_2 < 1$. The l.s. is based on algorithm 3.5 of Nocedal & Wright (2006) (using $c_1 = 10^{-4}$ and $c_2 = 0.9$). This is a sophisticated l.s. requiring several evaluations of $E$ to approximate it via a cubic interpolant and a procedure to zoom in and out of the search interval.

**Backtracking**  This satisfies only that the step decreases $E$. It is much simpler and faster than the previous l.s., since all it does is, starting with $\alpha_k = 1$, multiply $\alpha_k$ by a constant factor $\rho \in (0, 1)$ until a step size is found such that $E(\mathbf{x}_k + \alpha_k \mathbf{s}_k) < E(\mathbf{x}_k)$. We used $\rho = 0.8$.

**Constant**  We use a constant step size $\alpha_k = 1/L$ where $L$ is the Lipschitz constant for $E$. This is the simplest and fastest l.s., since no step sizes are searched. With gradient descent, it ensures that $E$ decreases at each iteration and that we converge linearly to $\mathbf{x}^*$. In the example of fig. 5, $L = 4.511$ so $\alpha_k = 0.221$. We use it only with gradient descent, since for other methods it is difficult to find a constant step size that is guaranteed to decrease $E$.

Fig. 5 compares the line searches for gradient descent ($\mathbf{s}_k = -\nabla E(\mathbf{x}_k)$). Clearly, gradient descent is best with a l.s. that ensures the Wolfe conditions hold; it converged within 200 iterations while the other two did not reach the desired accuracy in 1 000 iterations. The extra time spent in the l.s. compensates by reducing drastically the number of iterations. One reason why this happens is that the value of the step size $\alpha_k$ can vary significantly across iterations.

Fig. 6–7 compare the line searches for Newton's method ($\mathbf{s}_k = -\nabla^2 E(\mathbf{x}_k)^{-1} \nabla E(\mathbf{x}_k)$). In terms of the number of iterations required to reach the desired accuracy, the Wolfe l.s. is consistently better than backtracking but by a minimal margin, it just saves a few iterations (1 to 2 typically). Note that, from fig. 6 (see also the main text), Newton's method takes usually less than 10 to converge, rarely reaching 14, so there is very little room to improve that. In runtime, the Wolfe l.s. is around twice slower than backtracking (1.5–5× slower depending on the case), and never faster than backtracking. This is a consequence of the Wolfe l.s. taking more time per iteration, particularly given that each backtracking l.s. tries very few steps before succeeding (see fig. 3), very often just one. Although the precise runtimes will depend on the implementation of the Wolfe l.s. (we used Matlab), it is clear that backtracking is likely faster and certainly simpler. Also, a good l.s. for Newton's method must ensure that step sizes of 1 are used once near enough the minimizer to ensure quadratic convergence.

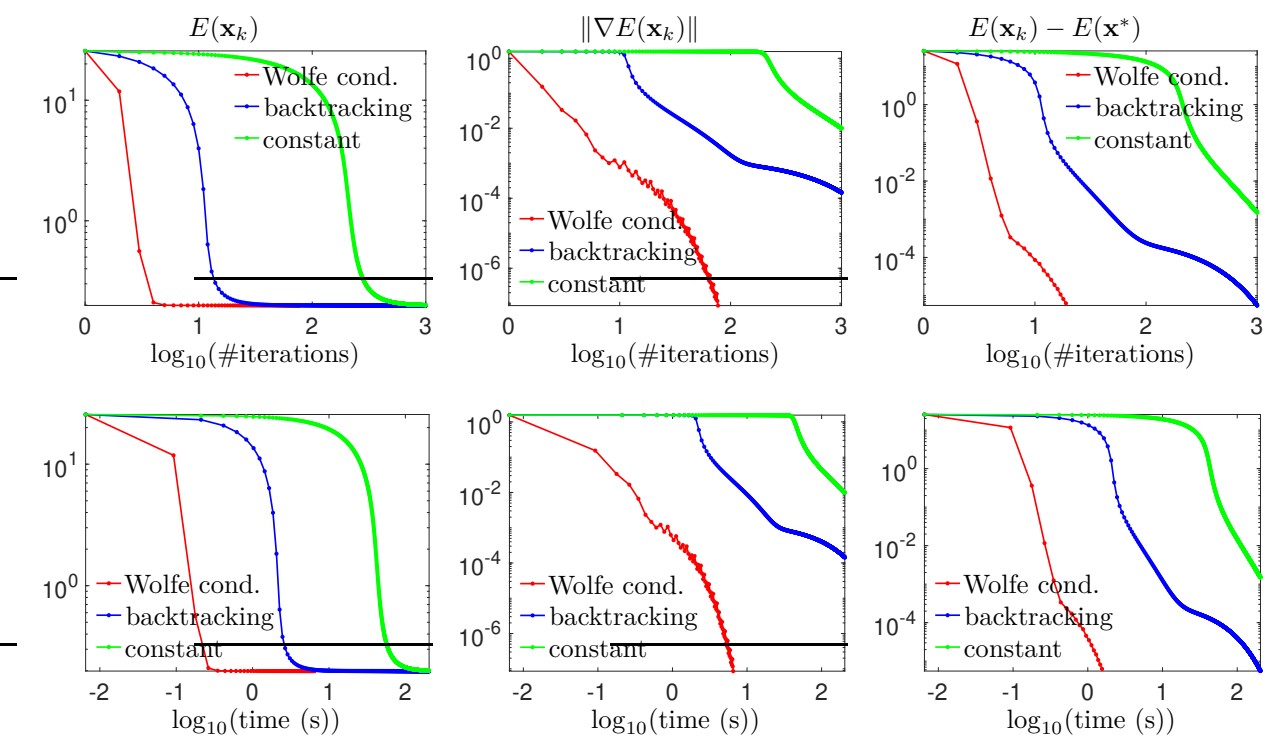

Figure 5: Types of line search in gradient descent: satisfying the Wolfe conditions, backtracking and using a constant step size $1/L$ (where $L$ is the Lipschitz constant of the objective function $E$). The problem uses a source instance from the VGGFeat256 dataset with $\lambda = 10^{-3}$. In each column we plot $E(\mathbf{x}_k)$, $\|\nabla E(\mathbf{x}_k)\|$ and $E(\mathbf{x}_k) - E(\mathbf{x}^*)$ (where $\mathbf{x}^*$ was estimated using Newton's method), for each iterate $k = 1, 2, \ldots$ Row 1 shows the number of iterations in the X axis and row 2 the runtime (s). Gradient descent was run with each line search until either $\|\nabla E(\mathbf{x}_k)\| < 10^{-7}$ or $1\,000$ iterations were reached.

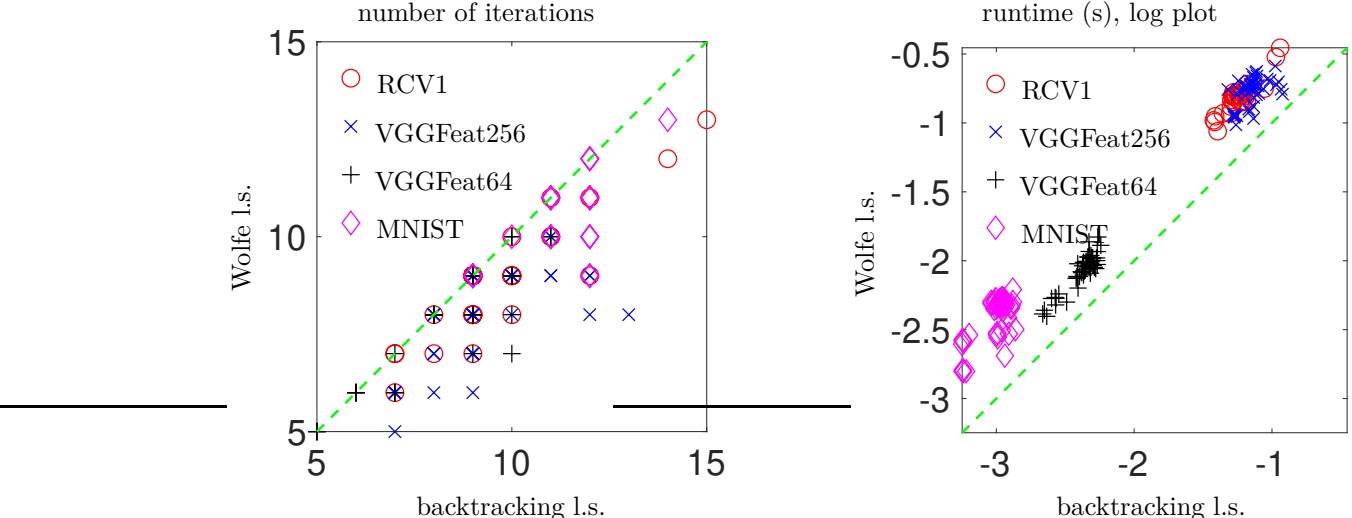

Figure 6: Types of line search in Newton's method: satisfying the Wolfe conditions and backtracking. We plot the number of iterations and runtime (s) over a collection of 50 problem instances for each of 4 datasets (these are the same problems as in fig. 1).

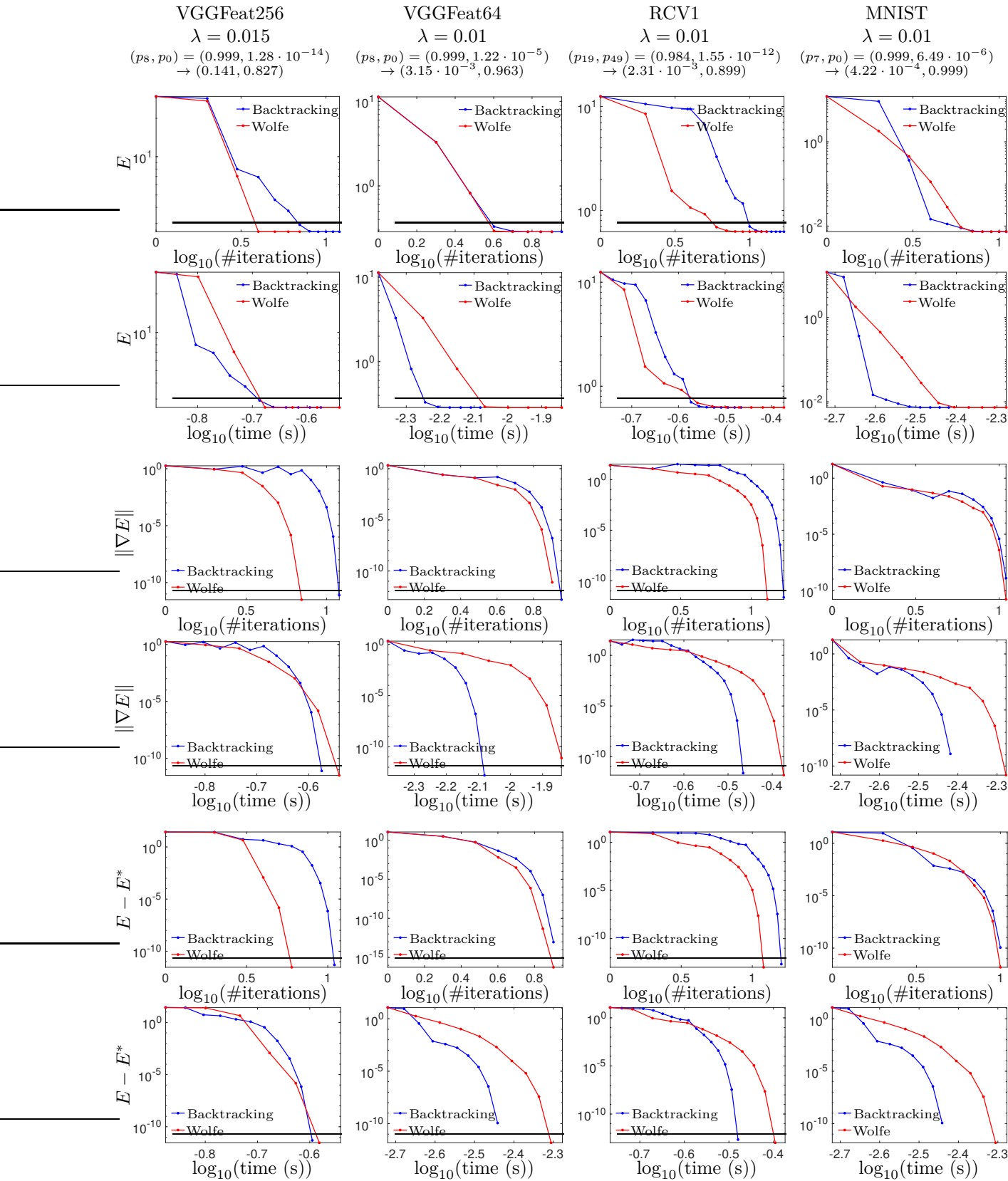

Figure 7: Types of line search in Newton's method: satisfying the Wolfe conditions and backtracking, as learning curves for one problem instance per dataset (columns), as in fig. 2.

# D  MNIST with 100 classes

We show another experiment using a large number of classes based on MNIST. In the original MNIST dataset, we have 10 digit classes (0 to 9) and each image is of 28×28. We can create a "double-digit" image by concatenating (side by side) two MNIST images, of classes $i$ and $j$, to generate an image of 28×56 of class $ij$ (say, a digit of class 6 and a digit of class 4 generate a digit of class 64). Thus we generate an "MNIST-100" dataset where the instance dimensionality is $D = 1\,568$ and the number of classes $K = 100$ (00 to 99). We constructed a training set of $N = 60$k double-digit images by creating 600 images for each class, each image being the concatenation of two random training set images of the corresponding class; likewise, we created a test set of 10k double-digit images. We trained a softmax classifier on the training set, which achieved 1.92% and 22.12% training and test error, respectively (this is likely overfitting, but it is fine for our purposes of setting up an inverse classification problem).

This way, we create a harder inverse classification problem, because we have more feature dimensions, many more classes and hence many more parameters, which increases the computation for each method. But it also makes the Hessian more ill-conditioned (less round), since from theorem 3.3 the Hessian has $K - 1$ eigenvalues greater or equal than $\lambda$ and the rest $(D - K + 1)$ equal to $\lambda$.

Fig. 8 shows the learning curves for a source instance of each dataset. Note that Newton's method barely needs more iterations to solve the MNIST-100 problem than to solve the MNIST one (around 10 in both cases) while all the other methods take twice as many iterations as before or more. The only exception is BFGS, which now takes fewer iterations than before; this makes sense, since after Newton's method it is the method that most uses second-order information. In terms of runtime, Newton's method is again the fastest by far, with a bigger gain than in MNIST. BFGS becomes the slowest method because of its costly matrix-vector multiplications with a $D \times D$ matrix.

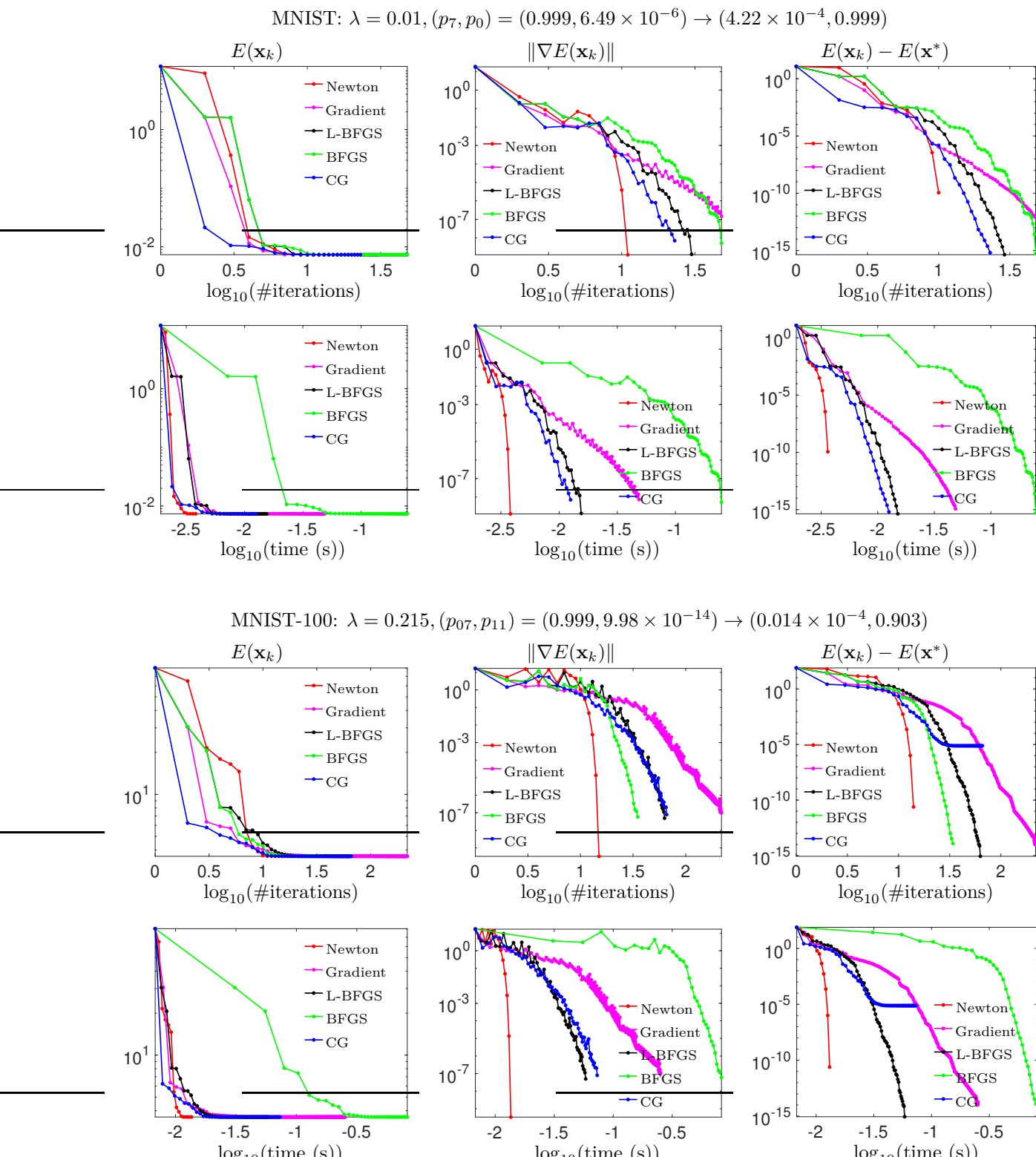

Figure 8: Learning curves in a problem instance from the MNIST (top) and MNIST-100 (bottom) datasets. In each column we plot $E(\mathbf{x}_k)$, $\|\nabla E(\mathbf{x}_k)\|$ and $E(\mathbf{x}_k) - E(\mathbf{x}^*)$ (where $\mathbf{x}^*$ was estimated using Newton's method), for each iterate $k = 1, 2, \dots$ For each dataset, row 1 shows the number of iterations in the X axis and row 2 the runtime (s).

