# OpenReview forum: "Inverse classification with logistic and softmax classifiers: efficient optimization"
_TMLR — Accepted by TMLR_

### Review · Reviewer_TYP4 · 2025-11-22

**Summary Of Contributions:**

Inspired by the growing interest in a class of optimization problems known as “inverse classification” (e.g., counterfactual explanations and adversarial examples), the authors investigated the speed of obtaining solutions for these problems in real-time or interactive scenarios.

Specifically, the authors theoretically examined how to solve these optimization problems efficiently for two commonly used classifiers: logistic regression and SoftMax classification. They show that, in the logistic regression setting, optimization for inverse classification problems can be solved in closed form. For SoftMax classification, although a closed-form solution is not available, the authors show that the problem can be solved efficiently through an iterative approach using Newton’s method combined with a suitably reformulated Hessian.

The paper concludes with experimental results across multiple datasets, providing empirical support for the theoretical findings.

**Additional Comments:**

Read the requested changes above.

**Audience:**

Yes

**Audience Explanation:**

The reviewer is convinced some individuals in the TMLR's audience would be interested in knowing the findings of this paper.

**Broader Impact Concerns:**

The paper does not pose any broader ethical concern.

**Claims And Evidence:**

Yes

**Claims Explanation:**

To the best of the reviewer's knowledge, the claims made in the submission are accurate, convincing, and well-supported. The theoretical arguments appear credible, and the empirical results provided support the theoretical claims.

**Requested Changes:**

The reviewer is curious whether the problem addressed in this paper can be empirically extended to any dataset, particularly in the K-class SoftMax setting. If not, the authors are encouraged to include in the revised version more information about the datasets used and any reasons the approach may fail on others.

Secondly, the authors are encouraged to extend their experiments to the CIFAR-10 and/or SVHN datasets.

---

> ### Author Response · Authors · 2026-01-08
>
> Thank you for your comments. To address your questions:
>
> - *[...] whether the problem addressed in this paper can be empirically extended to any dataset, particularly in the K-class SoftMax setting [...] any reasons the approach may fail on [other datasets].*
>
>   Our approach will work correctly with any dataset with real-valued features, since our formulation and analysis depend only on the structure of the logistic regression or K-class softmax classifier and the L2 norm, and not on dataset-specific properties. Specifically, the problem is strongly convex and has a unique solution, to which which Newton iterations converge quadratically while requiring a KxK inverse (rather than DxD) in the softmax case, and with a closed-form solution in the logistic regression case.
>
>   The choice of datasets was intended to illustrate different cases of K (binary or multiclass), feature dimensionality D (from medium to very high) and data type (image pixels, neural network image features and documents).
>
> - *the authors are encouraged to extend their experiments to the CIFAR-10 and/or SVHN datasets.*
>
>   Our experiments already cover comparable and more challenging settings than those of CIFAR-10 or SVHN. In particular, MNIST and the VGGFeat64/VGGFeat256 datasets are derived from real images, with the VGG-based features having significantly higher dimensionality than CIFAR-10, as well as more classes. Given the fact that we can thoroughly characterize the theoretical aspects of the problem (convergence rate to the unique solution and computational complexity of the iterations), we think that our experiments validate the approach in challenging cases. Thus we'd request the reviewer to reconsider the need for additional experiments.

---

### Review · Reviewer_ySbN · 2025-11-24

**Summary Of Contributions:**

The paper studies inverse classification for fixed logistic and softmax models. For softmax classifiers, the paper demonstrates that the Hessian has a structure that enables extremely efficient Newton steps via a K × K system. For logistic regression, the problem reduces to solving a single scalar equation, yielding fast solutions. Experiments on various datasets (MNIST, RCV1, VGG) demonstrate speed advantages over gradient-based and Newton methods.

**Additional Comments:**

The paper is clear, rigorous, and well-organized. The contribution is valuable, and the experimentations are sound. I think this is a good paper for TMLR.

**Audience:**

Yes

**Audience Explanation:**

The work will be of interest to researchers in optimization for explainability, adversarial examples, and counterfactuals, as well as those developing efficient proximal or Newton-type solvers. The scope is quite narrow, but the paper is well crafted, and its contributions are highly relevant.

**Broader Impact Concerns:**

No major concerns. The paper touches on adversarial examples and counterfactuals. A brief note could emphasize dual-use considerations (interpretability vs. adversarial use), but no deeper ethical issues arise.

**Claims And Evidence:**

Yes

**Claims Explanation:**

The theoretical claims are supported by mathematical derivations that appear to be correct and justify the method. The experiments are extensive and consistent with the theoretical insights. The claims focus on optimization efficiency, and the experimental evidence supports these claims.

**Requested Changes:**

* As a reader not extremely familiar with the literature on proximal operators, I am unable to directly identify which parts of the mathematical derivations are new and which ones are well-known in this literature. Therefore, I think the paper could be better anchored in the global optimization literature to highlight connections and differences.
* I would appreciate a few discussions on the application side of the problem, what solutions the inverse optimization problem (counterfactual explanations) produces, and what their main characteristics are. This will allow the reader to better position the paper in terms of direct application use.

---

> ### Author Response · Authors · 2026-01-08
>
> Thank you for your comments. To address your questions:
>
> - *As a reader not extremely familiar with the literature on proximal operators, I am unable to directly identify which parts of the mathematical derivations are new and which ones are well-known in this literature. Therefore, I think the paper could be better anchored in the global optimization literature to highlight connections and differences.*
>
>   We don't use any ideas of proximal operators. We apply tools of nonlinear optimization, as well as the SMW formula, to a specific case: K-class softmax classifier or logistic regression with the L2 norm. The mathematical structure of this case affords important computational advantages which do not arise with other classifiers or other norms. Theorems 3.1 and 3.2 follow directly from convex analysis, but the remaining theorems exploit the said structure. The corresponding results are, as far as we know, new (the low-dimensional Hessian representation for K-class softmax and the closed-form solution for logistic regression).
>
> - *[...] a few discussions on the application side of the problem, what solutions the inverse optimization problem (counterfactual explanations) produces, and what their main characteristics are.*
>
>   The inverse optimization problem produces counterfactual examples that answer a simple question: what is the smallest change to the input that would make the model change its prediction to a desired class? When considering a generic classifier and norm, this problem can be hard (local optima, non-differentiability, etc.). We focus on a particular case of classifier and norm which is highly practical and for which the solution (i.e., counterfactual explanation) is unique and, thanks to the problem structure, can be computed extremely efficiently.
>
>   This is particularly useful in real-time or interactive scenarios where fast feedback matters. For instance, in a credit approval or document moderation system, a user may want to understand which minimal changes to the input, such as adjusting an income-related feature or modifying a small portion of the text, would be sufficient to change the model's decision. The efficiency of our approach makes it possible to compute such counterfactual explanations on demand, allowing users or practitioners to explore different "what-if" scenarios interactively.

---

> > ### Comment · Reviewer_ySbN · 2026-01-12
> > **Thank you**
> >
> > Thank you for your responses. This addresses my questions.

---

### Review · Reviewer_CzjD · 2025-12-14

**Summary Of Contributions:**

In this paper the authors propose inverse classification as an optimization problem. The authors propose an objective for the inverse classification problem. The authors show theoretically 1. the convergence for a softmax classifier, 2. the close form solution for the logistic regression. Experimentally, the authors show on multiple datasets that the runtime is low.

**Audience:**

Yes

**Audience Explanation:**

The concept of inverse classification itself is interesting.

**Claims And Evidence:**

Yes

**Claims Explanation:**

The paper is well supported by theoretical evidence.

**Requested Changes:**

- The name inverse classification is a bit confusing. In fact, the optimization problem is not finding x given y and f right. It's finding other x given an initial x, y, and f.
- The current formulation (2) itself lacks practical motivation. It is unclear to me what is the motivation of putting a L2 distance bound between the target input and the source input in real world scenarios within an optimization objective. Note that even for traditional adversarial example, the L_infty ball is more commonly used compared to the L_2 ball. Therefore, the authors should add examples on why such input space L_2 similarity is important for practical use cases.
- Similar problem has been studied in adversarial robustness community and people have been using optimization methods such as BFGS, GD, Newton's Method to perform targeted adversarial attack. Convergence on such attacks has also been studied in a lot of prior works, on more generalized classifier. Therefore, the contribution seems a bit limited.
- The scope of experiments is also limited. It's hard to tell the practical motivation of using these TFIDF features of images. While I think the general idea of inverse classification is a good one, it needs real world data and use cases to justify its motivation of being practical.

---

> ### Author Response · Authors · 2026-01-08
>
> Thank you for your comments. To address your questions:
>
> - *The name inverse classification is a bit confusing. In fact, the optimization problem is not finding x given y and f right. It's finding other x given an initial x, y, and f.*
>
>   True, but still the problem is "inverse" in spirit, in that it goes in the opposite direction to classification. The latter is a "search" in y-space where x is provided; the former is a search in x-space where y (and an "initial" x) is provided.
>
> - *The current formulation (2) itself lacks practical motivation [...] even for traditional adversarial example, the L_infty ball is more commonly used compared to the L_2 ball.*
>
>   In general, the choice of norm depends on the application. In the adversarial examples literature, the L2 norm has been used in several influential works, such as "Intriguing properties of neural networks" (Szegedy et al., 2014) (section 4.1) or "DeepFool: a simple and accurate method to fool deep neural networks" (Moosavi-Dezfooli et al., 2016) (section 1). Other norms (L1 or L_infty) have also been used, as you note. However, the primary goal of our work was to exploit the mathematical structure that occurs in the combination of L2 norm with softmax or logistic regression classifiers, which affords an extremely efficient solution. We are not aware of a similarly efficient solution for other norms.
>
> - *[...] in adversarial robustness community [...] people have been using optimization methods such as BFGS, GD, Newton's Method to perform targeted adversarial attack. Convergence on such attacks has also been studied in a lot of prior works, on more generalized classifier. Therefore, the contribution seems a bit limited.*
>
>   Indeed, much work has used various optimization methods, but, as you note, without making specific assumptions about the classifier, without exploiting model and norm-specific structure. Our contribution focuses on the mathematical structure of a particular but important formulation. This allows us to characterize thoroughly the theoretical aspects of the problem (quadratic convergence rate to the unique solution and computational complexity of the iterations, or even closed-form solution) and provide an extremely efficiently computation, achieving runtimes ranging from milliseconds to around a second even for very high-dimensional instances and many classes.
>
> - *The scope of experiments is also limited. It's hard to tell the practical motivation of using these TFIDF features of images. While I think the general idea of inverse classification is a good one, it needs real world data and use cases to justify its motivation of being practical.*
>
>   We use TF-IDF features only for the document dataset (RCV1). The image datasets (MNIST, VGGFeat64, and VGGFeat256) use pixel-based features or deep neural network features, not TF-IDF features.
>
>   All datasets used in our experiments represent real-world data and practical settings. MNIST, VGGFeat64, and VGGFeat256 are derived from real images, with the latter two using features extracted from a pre-trained VGG network on ImageNet. The RCV1 dataset consists of real newswire stories, which closely resemble real-world news articles and text classification tasks.

---

### Decision · Action_Editor_Wx1y · 2026-02-03

**Recommendation:** Accept with minor revision

**Additional Comments:**

Two of the reviewers recommended acceptance while one recommended rejection, the latter due to a remaining concern about the degree of overlap with adversarial example methods. On this last point, I agree with the authors that their contribution is distinguished by specializing to cases that have the mathematical structure needed to develop their efficient solutions. I note that the rebuttal also resolved (in my view) comments about TF-IDF features and additional datasets.

I recommend acceptance with minor revision to address a couple of small points based on the reviews (and because I did not see a revised version uploaded during the rebuttal period).
1. I think it would be good to comment more on the $\ell_2$ distance aspect, namely as a characteristic of the returned solutions (referring to Reviewer ySbN's comment) and as a limitation of the work (based on Reviewer CzjD's comments).
1. Following up on Reviewer ySbN's comment, the camera-ready version could indicate more clearly the transition in Section 3 between well-known results in convex analysis and exploitation of the particular mathematical structure.

**Audience:**

Yes

**Audience Explanation:**

Since inverse classification is a higher-level problem as mentioned above, the paper should be of interest to the communities that work on variants of inverse classification for finding adversarial examples, counterfactual explanations, etc. The paper might also be of interest to researchers working in second-order optimization methods.

**Claims And Evidence:**

Yes

**Claims Explanation:**

This submission first identifies inverse classification as a common, higher-level problem that has been tackled in different sub-areas: finding adversarial examples, counterfactual explanations, and model inversion. The main contribution is the development of highly efficient solutions for special cases where the classifier is logistic regression or a softmax classifier and the proximity of the returned instance to the source instance is measured using $\ell_2$ distance.

Reviewers found the main contribution to be well supported. On the theoretical side, the derivations were judged to be correct. For logistic regression, the efficiency is due to the availability of a closed-form solution. For softmax classifiers, it is shown that the structure of the Hessian allows for efficient Newton steps, leading to a fast Newton algorithm with quadratic convergence rate. On the experimental side, reviewers thought that the efficiency advantage was clearly and extensively demonstrated.